Appendicular anatomy of the artiopod Emeraldella brutoni from the middle Cambrian (Drumian) of western Utah

http://orcid.org/0000-0003-2256-1872 Lerosey-Aubril Rudy rudy_lerosey@fas.harvard.edu
http://orcid.org/0000-0002-6801-7373 Ortega-Hernández Javier jortegahernandez@fas.harvard.edu
Museum of Comparative Zoology and Department of Organismic and Evolutionary Biology, Harvard University , Cambridge, MA , USA
Lieberman Bruce
Electronic publication date: 2019 Oct 31
Publication date: 2019
Volume: 7
Electronic Location ID: e7945
Received 2019 Jul 29; Accepted 2019 Sep 24
Copyright: © 2019 Lerosey-Aubril and Ortega-Hernández
Copyright year: 2019
Copyright holder: Lerosey-Aubril and Ortega-Hernández
License: This is an open access article distributed under the terms of the Creative Commons Attribution License, which permits unrestricted use, distribution, reproduction and adaptation in any medium and for any purpose provided that it is properly attributed. For attribution, the original author(s), title, publication source (PeerJ) and either DOI or URL of the article must be cited.
License URL: https://creativecommons.org/licenses/by/4.0/

Keywords: Exceptional preservation, Miaolingian, Artiopoda, Vicissicaudata, Caudal flaps, Aglaspidida

Funding: The authors received no funding for this work.

==============================
The non-biomineralized artiopod Emeraldella brutoni Stein, Church & Robison, from the middle Cambrian (Drumian) Wheeler Formation in Utah represents the only confirmed occurrence of the genus Emeraldella outside of the stratigraphically older (Wuliuan) Burgess Shale Konservat-Lagerstätte in British Columbia. The hitherto known sole specimen of this species is preserved in dorsal view and lacks critical information on the ventral appendages. Here, we redescribe E. brutoni based on a new completely articulated specimen that illustrates the appendage organization in exceptional detail. The main body consists of a cephalic region covered by a semicircular head shield, a trunk including 10 tergites with expanded pleurae plus a cylindrical terminal segment, and a long articulated tailspine. The head carries a pair of elongate and flexible antennae, a pair of lateral eyes, and three pairs of post-antennal appendages. We report the presence of eyes in Emeraldella for the first time. The first post-antennal limb solely consists of an endopod with well-developed paired spiniform endites. The remaining cephalic appendages and those associated with all but the last trunk segments possess exopods terminating in paddle-shaped, distal lobes fringed with robust setae. The cylindrical terminal segment bears a pair of posteriorly oriented caudal flaps reminiscent of trunk exopods, and a styliform, possibly uniarticulate tailspine longer than the main body. The new data on E. brutoni suggests an exopodal origin for the paired caudal structures in Vicissicaudata, and improve our understanding of the fundamental organization of this major clade within Artiopoda.

Introduction

The House Range and the Drum Mountains of western Utah are internationally renowned for their richly fossiliferous Miaolingian (“middle Cambrian”) strata, and particularly the beautifully preserved trilobites they contain. The abundance of trilobite fossils motivated the opening of a first quarry in the Wheeler Amphitheatre (Wheeler Formation; House Range) in the 1960s (Robison, Babcock & Gunther, 2015). Others have followed since then, including quarries in the Marjum and Weeks formations, and subsequently the area has become a major destination for trilobite collectors around the world. The exploitation of the local federal lands by fossil dealers and amateur collectors searching for trilobites led to the realization that these three Miaolingian units also contain the remains of non-biomineralizing organisms (Robison, 1991; Lerosey-Aubril et al., 2018). At least four exceptionally-preserved biotas have been described from these strata to date, including the two Wheeler biotas of the House Range and the Drum Mountains (both early Drumian), the Marjum biota (upper Drumian), and the Weeks biota (Guzhangian). Except for some early reports (Brooks & Caster, 1956; Rigby, 1969), the scientific study of these exceptionally-preserved faunas really started in the mid-1970s and became important in the 1980s, with a series of seminal contributions by Rigby & Gutschick (1976), Rigby (1978, 1983), Willoughby & Robison (1979), Robison & Richards (1981), Conway Morris & Robison (1982, 1986, 1988), Robison (1984, 1985, 1990), and Briggs & Robison (1984). Most of the remarkable fossils described in these publications had been found by members of the general public, either avocational paleontologists (e.g., the Gunther family, T. Johnson, or G. Melloy, to name a few; Gunther & Gunther, 1981) or professional fossil collectors (e.g., R. Harris). This remains true of the materials described in more recent publications (Briggs et al., 2008; Lerosey-Aubril et al., 2012, 2013a; Conway Morris et al., 2015; Lerosey-Aubril, 2015; Ortega-Hernández et al., 2015), with the reason being that the soft-bodied fossils in those strata are particularly rare, especially in the Wheeler and Marjum formations. A total of 40 years after the beginning of their scientific explorations, the Cambrian Konservat-Lagerstätten of western Utah are not “minor” anymore, as once described by Conway Morris (1985). These deposits have yielded between 20 and 45 soft-bodied species each (Robison, Babcock & Gunther, 2015; Lerosey-Aubril et al., 2018) and therefore qualify as Tier 2 Burgess Shale-type deposits (Gaines, 2014). However, only a handful of the remarkable fossils found in those beds every year end up in public institutions, and consequently many of the non-biomineralizing taxa in the area are still officially known only from a few (usually incomplete) specimens. Our current restudy of the Wheeler biota from the House Range reveals that 20 of the 45 soft-bodied species it contains (44.4%) are described in the scientific literature based on a single fossil, and an additional nine species (another 20%) are known from only two or three specimens. In this context, any new soft-bodied fossils from these Konservat-Lagerstätten hold the promise for significantly improving our understanding of these exceptional Cambrian biotas. In this contribution we describe a new specimen of Emeraldella brutoni (Stein, Church & Robison, 2011) (Fig. 1), an artiopod species hitherto known from a single specimen found in the Wheeler strata of the Drum Mountains (Stein, Church & Robison, 2011). The new specimen is preserved in a different orientation from that of the type material, providing valuable insights into the as-yet unknown appendicular anatomy of this taxon.

Figure 1 Holotype (KUMIP 321500) of Emeraldella brutoni from the Cambrian (Drumian) Wheeler Formation in the Drum Mountains.

Pictures were taken with the specimen immersed in dilute ethanol under cross-polarized light (Credit: M. Stein). (A) General view of the counterpart (mirrored). (B) General view of part. (C) Detail of anterior cephalic region of counterpart. (D) Detail of caudal flaps of part. Abbreviations: an, antenna; ar, anterior region of headshield; cf, caudal flap; cs, cylindrical terminal segment; hs, head shield; ms, marginal setae; Tn, trunk tergite n; ts, tailspine.

Geological setting

The four Cambrian exceptionally-preserved biotas of western Utah are found in the Miaolingian succession exposed in the House Range and the Drum Mountains, a few tens of kilometers east and north of Delta (Robison, 1991; Robison, Babcock & Gunther, 2015; Lerosey-Aubril et al., 2018). These shale-dominated deep-water deposits are at odds with the otherwise carbonate-dominated, wave-influenced sedimentological context characterizing the Miaolingian Series in the Great Desert of Utah (Miller, Evans & Dattilo, 2012; Foster & Gaines, 2016). This is explained by the development locally of a fault-controlled, deep-water basin during the terminal Wuliuan, which formed a prominent re-entrant into the seaward margin of the carbonate platform rimming the continent in the Early Paleozoic (Rees, 1986; Miller, Evans & Dattilo, 2012; Foster & Gaines, 2016), where fine-grained siliciclastic sediments of the outer detrital lithofacies belt were deposited. This asymmetrical trough, the so-called House Range Embayment (Rees, 1986), was subsequently filled during the Drumian to Guzhangian times by the deposits of the Wheeler, Marjum, and Weeks formations (in ascending stratigraphic order), which was accompanied by the reduction of its spatial extent. Shallow-water, carbonate platform settings were restored in the late Guzhangian (uppermost Weeks Formation). The Wheeler, Marjum, and Weeks formations form a continuous succession of up to 600 m in thickness, which is dominated by calcareous mudstone with subordinate shaly to flaggy limestone (Foster & Gaines, 2016).

The Wheeler Formation was deposited when the House Range Embayment had its broadest spatial extent. It mostly outcrops in the House Range and the Drum Mountains, where it has yielded two clearly distinct exceptionally-preserved biotas of lower Drumian age (Ptychagnostus atavus Biozone) (Robison, Babcock & Gunther, 2015; Lerosey-Aubril & Skabelund, 2018). The new fossil of E. brutoni described here was recovered from Wheeler strata in the Drum Mountains, as was the holotype and previously only known specimen of this species (Stein, Church & Robison, 2011). There, the Wheeler Formation is twice as thick (c. 300 m) and more calcareous than in the House Range (Brett et al., 2009), which is generally interpreted as evidence of its more proximal position on the shelf ramp forming the eastern (now northern) margin of the embayment.

Materials and Methods

The new material of E. brutoni consists of the part and counterpart of a single complete individual compressed in lateral view, which is deposited in the Natural History Museum of Utah (UMNH.IP.6162a and b, respectively; Fig. 2). This specimen was found by B. Sisson as a loose block at a Wheeler locality well-known to fossil collectors (GPS: 39°30′13"N 112°59′23"W). This site is one of the two sections that has yielded most of the soft-bodied components of the Wheeler biota of the Drum Mountains (“section c” of Halgedahl et al., 2009). This remarkable biota is composed of 71 species, including 30 species of non-biomineralizing organisms (Robison, Babcock & Gunther, 2015).

Figure 2 New specimen of Emeraldella brutoni from the Cambrian (Drumian) Wheeler Formation in the Drum Mountains.

Pictures taken with the specimen dry under cross-polarized light. (A) Part, UMNH.IP.6162a. (B) Counterpart, UMNH.IP.6162b (mirrored). Abbreviations: an, antenna; cf, caudal flap; cn, cephalic appendage n; cs, cylindrical terminal segment; hs, head shield; Tn, tergite n; trn, trunk appendage n; ts, tailspine.

Digital photographs were produced with a Nikon D5500 DSLR fitted with a Nikon 40 mm DX Micro-Nikkor lens. The specimen was photographed under direct light or cross polarized illumination, either dry or immersed in water. Series of images were taken with manual focusing at different focal planes, and subsequently stacked and assembled in Adobe Photoshop CS6. Schematic diagrams were produced in Inkscape.

Results

Systematic palaeontology

Phylum Euarthropoda Lankester, 1904

Subphylum Artiopoda Hou & Bergström, 1997

Superclass Vicissicaudata Ortega-Hernández, Legg & Braddy, 2013

Remark: We follow Lerosey-Aubril, Zhu & Ortega-Hernández (2017a) in considering Artiopoda as a subphylum, and Vicissicaudata as a superclass within it. Genus Emeraldella Walcott, 1912

Type species: Emeraldella brocki Walcott, 1912, from the middle Cambrian (Miaolingian Series, Wuliuan Stage) Burgess Shale in British Columbia (see also Bruton & Whittington, 1983; Stein & Selden, 2012).

Emended diagnosis: Artiopod with head containing one antennal and three limb-bearing post-antennal segments; antennae long, exceeding 80 articles in some species; first post-antennal limb consists of an endopod only, with reduced number of podomeres and well-developed spiniform endites. Trunk includes 10–11 tergites with well-developed pleural regions, and an elongate cylindrical terminal (“pre-telsonic”) segment bearing a pair of ventral caudal flaps and the tailspine. Proportions of endopod podomeres vary strongly along body. Endopod curving outward proximally, then downward at short, knee-like, fifth podomere. Sixth podomere long, distinctly stenopodous. Exopod tripartite, its proximal and middle parts articulating with basipod and first podomere, respectively; proximal part bearing lamellae, while middle and distal parts fringed with setae. Tailspine styliform, jointed, longer than the trunk, and lateroventrally flanked by exopods modified into caudal flaps.

Remarks: We propose minor updates to the emended diagnosis of Emeraldella by Stein, Church & Robison (2011) that emphasize the unique appendicular organization of this genus, as revealed by UMNH.IP.6162.Emeraldella brutoni Stein, Church & Robison, 2011

Figures 2–5

Figure 3 New specimen of Emeraldella brutoni from the Cambrian (Drumian) Wheeler Formation in the Drum Mountains.

Pictures taken with the specimen dry (A, E) or immersed in dilute ethanol (B, D) under cross-polarized light. (A, B) Part, UMNH.IP.6162a. (C) Composite interpretative drawing, combining details of both part and counterpart. (D, E) Counterpart, UMNH.IP.6162b (mirrored). Abbreviations: an, antenna; cf, caudal flap; cn; cephalic appendage n; cs, cylindrical terminal segment; ey, eye; hs, head shield; Tn, tergite n; trn, trunk appendage n; ts, tailspine.

Figure 4 New specimen of Emeraldella brutoni from the Cambrian (Drumian) Wheeler Formation in the Drum Mountains.

Pictures taken with the specimen dry (E, F) or immersed in dilute ethanol (A, B, D) under cross-polarized light. (A, B, D) Detailed views of the cephalic region (A), the eyes (B), and a first post-antennal appendage (D) in part (UMNH.IP.6162a). (C) Interpretative drawing of B. (E, F) Detailed views of the posterior trunk exopods and caudal flaps (E) and tailspine (F) in counterpart (UMNH.IP.6162b); note the articulation (arrowhead) associated with a tiny spine dorsally. Abbreviations: an, antenna; as, articulation spine; cf, caudal flap; cn, cephalic appendage n; en, endite; ey, eye; hs, head shield; ms, marginal setae; trn, trunk appendage n.

Figure 5 Morphological reconstruction of Emeraldella brutoni.

(A) Lateral view. (B) Dorsal view. (C) Ventral view. The morphology of the hypostome, sternites, tripartite exopod organization, and all but the first set of endopods are tentatively extrapolated from the better-known Emeraldella brocki from the Burgess Shale (see Stein & Selden, 2012). The presence of simple rather than tripartite endopod tips is generalized from the morphology of the well-preserved first post-antennal appendages.

Emended diagnosis: Species of Emeraldella with 10 trunk tergites with well-developed pleurae, lateral eyes, and robust setae on exopods and caudal flaps. Emended from Stein, Church & Robison (2011).

Material, locality, horizon: The type material consists of the holotype KUMIP 321500 (Fig. 1)—the part and counterpart of a complete dorsal exoskeleton associated with a few appendicular remains—which is housed in the Kansas University Museum of Natural History. The fossil was collected in the Drumian strata (Ptychagnostus atavus Biozone) of the uppermost part of the Wheeler Formation in the Drum Mountains, Millard County, Utah (see Stein, Church & Robison, 2011, for details on the locality). The new specimen (UMNH.IP.6162; Fig. 2) comes from the same strata a few tens meters to the east and is housed in the Natural History Museum of Utah.

Description of the new material: UMNH.IP.6162 is a completely articulated individual preserved as a lateral compression (Fig. 2), with a maximum main body length of 50 mm (sag.; tailspine excluded). Body consists of head region covered by head shield, 11-segmented trunk, and tailspine. Head shield longer (antero-posteriorly) than high (dorso-ventrally)—its dorsal margin almost flat posteriorly, strongly sloped down anteriorly—with straight posterior margin (Fig. 3); no dorsal ecdysial sutures or differentiated axis (Figs. 2–4A). Trunk segments 1–10 covered dorsally by broadly overlapping tergites of subequal length (sag.), and bearing ventral appendages (Figs. 2A and 3A–3C); terminal (11th) trunk segment cylindrical and bearing caudal flaps and tailspine (Figs. 2, 3, 4E and 4F). Tailspine spiniform, exceeding main body length (sag.), apparently uniarticulated, and with broad rounded base telescoped within posterior half of 11th trunk segment; single visible articulation at boundary between first and second thirds of tailspine, associated with short dorsal spine-like extension of proximal part (Figs. 2 and 4F).

Ventral structures preserved in UMNH.IP.6162 include one lateral eye (at least), a pair of antennae, the remains of post-antennal appendages, and a pair of caudal flaps (Figs. 2–4). Large ovoid patch of carbonaceous material on UMNH.IP.6162a interpreted as an eye; it is located under anterior third of headshield, close to its ventral margin, and partly overlaps another ovoid structure still largely covered by cuticle that likely represents the corresponding left eye (Figs. 4A–4C). Antennae flagelliform, elongate, and flexible (Figs. 2–4A); one antenna inserts on ventral surface of head a short distance below right eye, and projects antero-ventrally under head shield, then antero-dorsally, and finally abruptly bends dorsally; other antenna gently curves ventrally, then abruptly bends dorsally distally (Fig. 4A). First post-antennal appendages consist of elongate endopods only; they are equipped with strong, medially-directed, spiniform endites, and prominent terminal claws (one each), and comprise at least five podomeres (claw included; Figs. 4A and 4D). Following tergites associated with the appendages of one side only. Second post-antennal appendage represented by its exopod only; proximal anatomy unclear, but exopod extends into long paddle-shaped lobe fringed with robust setae distally (Figs. 3C–3E and 4E); corresponding endopod most likely concealed by overlying exopod and/or sediment. Last cephalic appendage and following ten trunk ones similar in preservation and morphology to second post-antennal appendage, except for moderate decrease of exopod length beyond mid-trunk (Figs. 3 and 4E). Terminal (11th) trunk segment bears pair of posteriorly oriented caudal flaps that apparently insert under its posterior half, as suggested by gap between them and exopod of preceding segment; caudal flaps resemble elongate trunk exopods with distal lobes bearing robust marginal setae (Figs. 3 and 4E). Discontinuous dark trace, likely made of carbon film, runs along ventral margin of body and produces short projections within exopod proximal parts (Fig. 3); this trace starts close to insertion sites of antennae and can be tentatively followed posteriorly to base of tailspine (Fig. 3C). Whether these structures represent remains of gut and digestive glands, circulatory system, decay fluids, or combination of two or more of these elements is unclear.

Remarks: UMNH.IP.6162 ideally complements the holotype and previously only specimen of E. brutoni, which is dorso-ventrally flattened and contains little appendicular data (Figs. 1 and 2). The new specimen confirms the presence of 10 trunk tergites with expanded pleurae in this taxon, and also illustrates other features that are unique within the genus. We propose to regard the presence of ventrally projecting lateral eyes, and robust setae on the exopods and caudal flaps as additional diagnostic characters of E. brutoni (Fig. 5). Stein, Church & Robison (2011; figs. 2, 3) interpreted several transverse marks on the proximal part of the tailspine of the holotype of E. brutoni as putative articulations; only one joint associated with a tiny dorsal spine can be observed in the new specimen, but the absence of other articulations might be an artefact of preservation, for the segmentation of the antennae is not preserved either.

Discussion

Comparison between Emeraldella species

The morphology of E. brutoni from the Wheeler Formation (Stein, Church & Robison, 2011), as informed by new specimen UMNH.IP.6162, shows close parallels with that of E. brocki from the Burgess Shale (Bruton & Whittington, 1983; Stein & Selden, 2012). Both taxa share a similar dorsal exoskeleton consisting of a semicircular head shield with acute genal angles, a series of freely articulating trunk tergites of subequal length (sag.) with expanded pleurae, a cylindrical terminal segment, and a spiniform tailspine exceeding the length (sag.) of the main body (Figs. 1, 2 and 5). Emeraldella species also possess a comparable appendage organization including: long antennae; first post-antennal limbs solely composed of spine-bearing endopods; following limbs with distal paddle-shaped exopodal lobes bearing marginal setae; and paired caudal flaps with marginal setae that occupy a lateroventral position relative to the tailspine (Figs. 1D and 4E).

Both E. brocki and E. brutoni have been suggested to possess an anterior sclerite, also referred to as “rostral plate” or “pre-hypostomal sclerite” (Stein, Church & Robison, 2011; Stein & Selden, 2012). However, tentative evidence for this sclerite has only been observed in a single specimen for each species (see Stein & Selden, 2012, fig. 4A; Stein, Church & Robison, 2011, fig. 2.1 and 2.2), while other well-preserved specimens of E. brocki that are exposed in dorsal or ventral views do not show any indications of this structure. UMNH.IP.6162 does not show traces of an anterior sclerite either, or any other additional structures associated with the head shield (Fig. 4A). What has been interpreted as a possible “pre-hypostomal sclerite” in the holotype seems in continuity with the rest of the headshield (i.e., no furrow; Stein, Church & Robison, 2011, fig. 2.1), and therefore likely corresponds to the slightly deformed anteromedial part of this sclerite (Fig. 1C). In the case of E. brocki (see Stein & Selden, 2012, fig. 4A), we believe that the putative anterior sclerite similarly represents a compression line on the anterior edge of the head shield, which has been produced by the outline of the adjacent antennae and first post-antennal limbs, rather than a discrete exoskeletal plate. The anterior sclerite of Cambrian euarthropods is usually clearly distinguished at the anterior edge of the head shield, and either accommodated in a median notch (Ortega-Hernández, 2015) or articulated with a natant hypostome (see Hou & Bergström, 1997; Edgecombe & Ramsköld, 1999). This is not observed in any of the two Emeraldella species and accordingly, we consider that there is no conclusive evidence supporting the presence of an anterior sclerite in this genus.

Stein, Church & Robison (2011) noted that the main feature that distinguishes E. brutoni from E. brocki is the presence of 10, rather than 11 trunk tergites with developed pleurae (Bruton & Whittington, 1983; Stein & Selden, 2012). UMNH.IP.6162 reveals further differences between the two species. E. brutoni features a set of lateral eyes (Figs. 4B and 4C), whereas ocular structures are confidently absent in the type species given the abundance of well-preserved material (Bruton & Whittington, 1983; Stein & Selden, 2012). The structures borne by the appendages of E. brutoni also appear more robust than those of E. brocki. For example, the first post-antennal endopod of E. brutoni are equipped with stout spines (Fig. 4C), rather than hair-like endites as in E. brocki (Stein & Selden, 2012). Likewise, the distal exopodal lobes and the caudal flaps bear closely-packed, lamella-like elements in E. brutoni (Fig. 4E), rather than delicate, widely-spaced marginal setae in the Burgess Shale species (Stein & Selden, 2012, figs. 4, 5, 7, 8, 10).

One of the most striking characteristics of E. brocki is the presence of extremely long antennae that can bear in excess of 80 articles in some specimens (Bruton & Whittington, 1983; Stein & Selden, 2012). The antennae of E. brutoni might have been just as long, but their preservation in the two available specimens does not permit to estimate their full extent, or even the number of articles composing their preserved parts (Figs. 1–4A). Most endopods (except for the first post-antennal pair) of UMNH.IP.6162 are not observable, which might be due to them being in a deeper topographic plane within the matrix or simply concealed by the exopods. If the latter case is true, this would indicate a significant difference in endopod length between the two species, as the endopods on the anterior half of the body of E. brocki extend well-beyond the dorsal exoskeletal boundaries, and are much longer than the corresponding exopods (Stein & Selden, 2012, figs. 5, 6, 7). Lastly, some aspects of the anatomy of E. brutoni—such as the shape of its hypostome, the morphology of its endopods (except for the first post-antennal ones) and proximal parts of its exopods, or the organization of its sternites—remain unknown, while they are well documented in E. brocki (see Stein & Selden, 2012). Gaining access to such anatomical details in the future might provide additional characters distinguishing the two species.

Origin of the caudal appendicular derivatives of Vicissicaudata

The affinities of Emeraldella within Artiopoda have become more or less stable in recent years, thanks to the availability of exceptionally preserved material and broad-scale phylogenetic analyses of Cambrian euarthropods (Edgecombe & Ramsköld, 1999; Cotton & Braddy, 2003; Stein & Selden, 2012; Legg, Sutton & Edgecombe, 2013; Ortega-Hernández, Legg & Braddy, 2013; Aria & Caron, 2017; Lerosey-Aubril, Zhu & Ortega-Hernández, 2017a). These analyses generally have recovered Emeraldella as forming a clade with aglaspidids, cheloniellids, and Sidneyia. This clade Vicissicaudata (Ortega-Hernández, Legg & Braddy, 2013), now regarded as a superclass (Lerosey-Aubril, Zhu & Ortega-Hernández, 2017a), is defined by a trunk with well-developed tergopleurae, except for at least one cylindrical terminal segment that also bears paired structures regarded by most workers as appendicular derivatives (Raymond, 1920; Hou & Bergström, 1997; Van Roy, 2005; Stein & Selden, 2012; Ortega-Hernández, Legg & Braddy, 2013). However, the morphology of the latter structures and their connection to the rest of the body vary greatly within the Vicissicaudata, and it is still unclear how each type has evolved from normal trunk appendages.

Aglaspidids are characterized by the presence of paired sclerotized plates covering ventrally the posteriormost two or three trunk tergites and the base of the tailspine (Hesselbo, 1992; Van Roy, 2005; Ortega-Hernández, Legg & Braddy, 2013). These so-called “post-ventral plates” have been repeatedly regarded as homologous to the caudal flaps of Emeraldella (Wills et al., 1998; Cotton & Braddy, 2003; Van Roy, 2005), despite substantial differences in shape, structure, and position between them. For a long time, the hypothesis of an appendicular origin of aglaspidid post-ventral plates heavily relied on the fact that these structures were paired (Cotton & Braddy, 2003; Van Roy, 2005), but recent studies have shed light on the origin of appendicular derivatives in this group. Indeed, Lerosey-Aubril et al. (2017b) showed that the broad-based tailspine characterizing many aglaspidids has likely evolved through the fusion of a cylindrical terminal segment and a needle-like telson. More importantly, the aglaspidid Glypharthrus trispinicaudatus demonstrates that such a freely articulated terminal segment also includes a pair of non-articulated caudal rami projecting from its posterior margin (Lerosey-Aubril, Zhu & Ortega-Hernández, 2017a), a condition closely reminiscent of Emeraldella and cheloniellids. It seems now likely that the “normal” appendages of the terminal segment of the vicissicaudatan ancestor initially evolved into caudal rami in the aglaspidid or aglaspidid + cheloniellid lineage. However, it remains unclear whether these caudal rami have later evolved into the post-ventral plates of more derived aglaspidids, or if the fusion of the pre-terminal segment and the telson has resulted in their loss; the latter scenario would imply that the post-ventral plates have evolved independently, possibly from a different body segment altogether (Lerosey-Aubril, Zhu & Ortega-Hernández, 2017a).

In cheloniellids, the terminal cylindrical trunk tergite bears a pair of rami, which can be short and articulated (Van Roy, 2006) or long and unsegmented (Stürmer & Bergström, 1978). Even if one accepts an appendicular origin for these structures, their derived morphology provides no clue as to how they have evolved from conventional trunk appendages, and their dorsal insertion (Van Roy, 2006) suggests evolutionary transformations unique to the cheloniellid lineage. Lastly, the exact nature of the uropods of Sidneyia is unclear. They are large, unsegmented structures (Bruton, 1981; Zacaï, Vannier & Lerosey-Aubril, 2016), which appear totally fused to the telson (Lerosey-Aubril, Zhu & Ortega-Hernández, 2017a). How exactly they might have evolved from regular trunk appendages remains unknown.

Of all known vicissicaudates, Emeraldella shows the least derived condition with regard to the structures borne by the cylindrical terminal trunk segment. Its caudal flaps provide a valuable reference point for understanding the origin of this diagnostic feature of the superclass. Most authors have considered an appendicular origin for the caudal flaps in E. brocki (Raymond, 1920; Hou & Bergström, 1997; Van Roy, 2005; Stein & Selden, 2012; Ortega-Hernández, Legg & Braddy, 2013), with the exception of Bruton & Whittington (1983), who regarded them as a single bilobate ventral plate. The well-preserved caudal flaps of E. brutoni (Figs. 1D and 4E) show a remarkable degree of similarity with the distal parts of the trunk exopods (Stein & Selden, 2012), including their lobate shape and presence of marginal setae. Interestingly, the main difference between the caudal flaps of the two Emeraldella species is also the main difference expressed by their trunk exopods, namely that both structures are fringed with delicate setae in E. brocki, whereas they bear robust, lamella-like ones in E. brutoni. Stein & Selden (2012) noted that the setae of a caudal flap are “shorter, more robust, and more widely spaced than those of the exopod” in E. brocki, but this does not seem to be the case in E. brutoni (Fig. 4E). The same authors reconstructed these flaps as bipartite, with short triangular proximal parts and elongate distal lobes (Stein & Selden, 2012, fig. 11A), but the morphology of the proximal region is mostly speculative, as acknowledged in their text (p. 16); unfortunately, UMNH.IP.6162 does not provide more information in this regard.

Based on these observations, we hypothesize that the evolution of the caudal flaps in Emeraldella likely resulted from a series of transformations of a typical biramous trunk appendage. These changes involved the loss of the endopod, the possible reduction or loss of the proximal exopod shaft, the elongation and narrowing of the exopod distal lobe, the re-orientation of this exopod from a ventral to a posterior direction, and possibly the migration of its insertion site on the body from the central region to the posterior region of the terminal segment. At least some of these transformations (re-orientation, posterior migration of insertion site) might have accompanied the evolution of the shape of the terminal cylindrical segment. In this evolutionary scenario, the caudal structures of other vicissicaudates, and particularly those of aglaspidids and cheloniellids, would fundamentally represent heavily modified exopods of the pre-terminal segment, which may find some support in the fact that none of them show evidence of biramy.

The new specimen of E. brutoni also points to another appendicular differentiation that could potentially further characterize vicissicaudates. UMNH.IP.6162 clearly shows that the first post-antennal appendage lacks an exopod in this species, demonstrating at the same time that the similar condition observed in E. brocki is legitimate, rather than a preservation artefact (Bruton & Whittington, 1983; contra Stein & Selden, 2012). Other vicissicaudates, such as Sidneyia (Bruton, 1981; Stein, 2013) and Cheloniellon calmani Broili (1932) (see also Stürmer & Bergström, 1978), are also known to lack exopods on at least the first post-antennal limb pair. No exopods whatsoever are known in the only known specimens of Aglaspis (Briggs, Bruton & Whittington, 1979) and Flobertia (Hesselbo, 1992) with appendicular remains (one specimen per taxon), but this might be due to incomplete preservation. Indeed, exopods are clearly expressed in the aglaspidid Kankhaspis from Siberia (Repina & Okuneva, 1969), although a comprehensive restudy of the material would be needed to determine whether all post-antennal cephalic appendages are biramous. Otherwise, appendicular data are missing for most aglaspidids (Hesselbo, 1992; Van Roy, 2005; Fortey & Rushton, 2009; Ortega-Hernández et al., 2010; Lerosey-Aubril et al., 2013a; Lerosey-Aubril, Ortega-Hernández & Zhu, 2013b; Ortega-Hernández, Legg & Braddy, 2013; Ortega-Hernández, Van Roy & Lerosey-Aubril, 2016; Lerosey-Aubril, Zhu & Ortega-Hernández, 2017a; Lerosey-Aubril et al., 2017b; Siveter et al., 2018) and cheloniellids (Chlupác, 1988; Dunlop, 2002), thus preventing a more complete assessment of how common the presence of a uniramous first post-antennal appendage is in vicissicaudates. Yet, particular attention should be given to this morphological trait in the future, for it might lead to a more accurate concept of Vicissicaudata.

Conclusions

The remarkable Miaolingian biotas of Utah have proved quite diverse with time, but most of their components remain poorly known due to the scarcity of available material. Each new soft-bodied fossil recovered from these beds can significantly improve our understanding of the anatomy or palaeocology of these taxa, and therefore refine our depiction of these ancient biotas. This is illustrated here with the description of a new specimen of E. brutoni, which allows the detailed description of the appendicular anatomy of this species for the first time. The appendage organization of E. brutoni suggest an exopodal origin for the caudal appendicular derivatives in the diverse clade Vicissicaudata. Its description also offers an opportunity to discuss the differentiation of the first post-antennal appendage into a uniramous structure (endopod) as a potentially diagnostic character of the superclass.

Further investigations on the clade Vicissicaudata might benefit from considering possibly related taxa, such as Molaria spinifera Walcott, 1912. Several studies have already argued for a possible sister-group relationship between Emeraldella and Molaria based on their tergite morphology, and presence of a multiarticulated tailspine (Whittington, 1981; Stein, Church & Robison, 2011; Legg, Sutton & Edgecombe, 2013). Molaria also shares with vicissicaudates the presence of a terminal cylindrical segment devoid of walking legs. This segment has been described as completely limbless (Whittington, 1981), but it bears a pair of short spines flanking the tailspine laterally, which might represent heavily modified appendages. Only a proper re-study of this taxon would permit to test this assumption.

Brock Sisson found the new specimen of E. brutoni described in this study. Carolyn Levitt-Bussian and Randal B. Irmis facilitated its study, and kindly assisted us during our visits to the Natural History Museum of Utah. The Bureau of Land Management, particularly Scott E. Foss and Greg McDonald, deposited the specimen in the museum and provided curation assistance. Martin Stein took the pictures of the holotype specimen and allowed us to use them in Fig. 1. A. Daley, G. Edgecombe, and an anonymous colleague provided us with detailed reviews that greatly improved our manuscript. We would like to express our sincere gratitude to all of these people for their invaluable help before and during the course of our study.

Additional Information and Declarations

Competing Interests

Author Contributions

Data Availability

The authors declare that they have no competing interests.

Rudy Lerosey-Aubril conceived and designed the experiments, performed the experiments, analyzed the data, contributed reagents/materials/analysis tools, prepared figures and/or tables, authored or reviewed drafts of the paper, approved the final draft.

Javier Ortega-Hernández conceived and designed the experiments, performed the experiments, analyzed the data, contributed reagents/materials/analysis tools, prepared figures and/or tables, authored or reviewed drafts of the paper, approved the final draft.

The following information was supplied regarding data availability:

The new material of Emeraldella brutoni consists of the part and counterpart of a single complete individual compressed in lateral view, which is deposited in the Natural History Museum of Utah (UMNH.IP.6162)

All relevant data is already provided in the manuscript.

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
