# Peer review of "Appendicular anatomy of the artiopod Emeraldella brutoni from the middle Cambrian (Drumian) of western Utah"

_PeerJ, doi:10.7717/peerj.7945_

## Round 0.1 · original submission · Minor Revisions

This is a high quality manuscript that can be an important and well cited paper in the journal, but there are some minor revisions that need to be made before publication. In particular, there are a number of very useful suggestions that the reviewers have provided in their quite detailed and positive reviews. I strongly encourage the authors to incorporate these into their revisions and with those the paper should be good to go.

·

Basic reporting

No comment (it's fine).

Experimental design

No comment (all fine).

Validity of the findings

No comment (in General Comments below I suggest a few details in the reconstruction that could be changed).

Additional comments

This is a useful update on a rare arthropod from the celebrated Wheeler Formation that was previously known from its holotype only. The description is very sound, and the interpretation of caudal flaps as exopod derivatives is of general interest to workers on Palaeozoic arthropods. I offer a few suggestions for the revision, and send along a handful of corrections, as follow

In Figure 4 (the reconstruction), I suggest showing a tripartite morphology of the exopod flaps, based on Stein and Selden’s interpretation of E. brocki, rather than non-articulated flaps. The tripartite morphology is included in the generic diagnosis in the paper, so the authors evidently infer that it applied to E. brutoni as well.

In the same figure (Figure 4), what is the evidence for the terminal claw on the endopods being simple? Bruton and Whittington (1983) presented compelling evidence for a complex claw in E. brocki, so given that the figures herein don’t obviously contradict that, one would be surprised were the claw in such a similar species to be so different.

In the abstract, indicate that the discovery of eyes in new for the genus.

Typos and such minutae:

Line 60: Perhaps use “J.K. Rigby” rather than “J. Rigby”, as he went by “Keith”.

Line 93: “at odds”, rather than “at odd”.

Line 107: “Drum Mountains”, rather than “Drums Mountains”.

Lines 134 and 135: parentheses are used for the authorships of Artiopoda and Vicissicaudata but not for Euarthropoda. This can’t be because the ranks have changed, as parentheses are used for taxon names only for Linnean binomials.

Line 178: “Lower” doesn’t make sense in the context of “Head shield slightly lower than long”. I am not sure what dimension “lower” refers to.

Lines 186 and 266: the relevant detail of the tailspine is in Fig. 2D (not 3D as cited).

Line 335: the caudal flaps are show in Fig. 2C (not 3C as cited). Please check the whole paper in case Figures 2 and 3 were reversed at some point and this wasn’t updated.

Line 364: The Aglaspis paper was Briggs et al., 1979 (not Briggs & Fortey, 1979 as cited).

Line 380: “The remarkable Miaolingian” sounds better than “the Miaolingian remarkable”.

Line 411: capitals for “Evolutionary Biology”.

Line 434: In this paper (as usual), it is “Donoghue, P.C.J.” rather than “Donoghue, P.C.”.

Line 478: “Quarterly”, rather than “Quaterly”.

Lines 481 and 500 (respectively): capital for “Communications” and “Reports”.

Figure captions: “tp” is kind of odd for “tailspine”. It implies the word breaks down to “tails pine” rather than “tail spine”.

·

Basic reporting

The manuscript submitted by Lerosey-Aubril and Ortega-Hernandez describes a new specimen of the taxon Emeraldella brutoni from the middle Cambrian Wheeler Formation of the USA, and discusses the importance of the new anatomical data in particular with regards to the trunk appendages. This is a well-written paper on an interesting fossil, and the introduction and background information provided are excellent for framing the importance of the study. The structure conforms well to PeerJ standards and is perfectly adapted for the discipline. The English is excellent throughout, with only a few minor suggestions listed in “General comments”. I commend the authors on this excellent manuscript and thorough description of the fossil. The figures are highly relevant and important, and they contain high quality photographs and an informative reconstruction of the fossil. The only relatively major suggestion for improvement is the addition of drawings of the fossil specimens, which are essential for adequately showing the details of the more complicated structures in the specimens (point 1 below). Otherwise my suggestions are relatively minor, but would help streamline and clarify certain parts of the study.

1. The manuscript contains beautiful photographs of the fossils, taken under all appropriate conditions for this type of material (dry and wet, incident and cross-polarised lighting). However, the lack of associated drawings of the specimens make it difficult to follow the descriptive text in some places. This is particularly true for the described details of: (1) the insertion of the antennae on lines 192-194; (2) the first post-antennae appendages on lines 194-197; (3) the second post-antennal appendage, in particular the slender proximal part. Please add drawings of the whole specimens with labelled features, and/or a closeup of the head region of Figure 3A.

2. In general, the written descriptive text needs to form a better link to the figures. The call-outs in the text never refer to labels in the figures, but they absolutely should. Every time a structure is mentioned or described, please as specific call-outs to, for example, “an in Fig. 1A”. Also, some of the figure call-outs don’t make sense for the text (for example line 209 where Figure 2D is a photo of an antennae but the text is describing something else). It is also clear that often the authors call-out to Figure 2 when they mean Figure 3, and vice versa (for example lines 189, 192, 197, 199, 203, 207, 211… and probably more) so please check all figure references again.

3. The manuscript contains an extremely complete and well-delivered systematic paleontology section, that conforms to typical format/structure for palaeontological taxonomy. There are only a couple small additions that would make their systematic section even clearer. This consists of (1) Adding the holotype number and museum where it is deposited somewhere around lines 169-170; (2) Adding the specimen number of the new specimen and its museum of deposition to the same section; (3) Adding a description of the second and third post-antennal cephalic limbs to the Emeraldella diagnosis somewhere around lines 147-148, or a statement that they are like the trunk appendages; (4) The Description section (lines 175-212) is written often in a style more similar to shortened diagnostic text, often achieved by removing small words such as “the”, “a”, “is”, etc, rather than fully-formed sentences. This makes it difficult to read at times, and could benefit from being written in a more fluid style with those smaller linking words being used to make full sentences. This is particularly evident near the start of the section, with the first paragraph (starting line 175) being written almost entirely in clipped sentences, but with more normal writing style creeping in more and more by the end of the section (lines 203 to 212 for example). (5) Minor spelling and grammatical errors listed in the “General comments".

4. The second (and to some degree third) post-antennal cephalic appendages look very different in the part (Figure 3A) than in the counterpart (Figure 2A, B). Is this just an artifact of preservation, and/or of the photographic settings used? This aspect could be cleared up if drawings of the specimens are included.

Experimental design

1. The experimental design is excellent and completely follows the appropriate approach of describing fossil material. The research question is very well defined and creatively written in the Introduction, effectively illustrating the gap in knowledge in this animal and emphasizing the importance of this single beautiful specimen. The rigorous investigation was performed to a high technical standard, although the inclusion of drawings of the fossil specimens would be highly appreciated (see my point 1 in “basic reporting”).

2. The only improvement in experimental design is quite minor. The figure captions talk about “composite images” in Figure 2A and 2B. What does this mean? Are they vertically stacked in order to get everything in focus? Are they serial horizontal images that have been stitched together? Please describe in the methods more clearly and specifically in what way the “digital images were processed” in Photoshop (line 128) and how this related to the composite images of Figure 2A and 2B.

Validity of the findings

The manuscript delivers a clear and coherent description of the significance and evolutionary implications of this new fossil specimen. The authors state clearly where they have more or less confidence in their conclusions and interpretations, which is excellent. The conclusions are exceptionally well stated and link nicely back to the introduction and context of the study. I have six points to mention here in regards to the interpretation of the fossil itself, of which only the first is rather important.

1. The eye: The authors describe a dark ovoid patch of carbonaceous material as a lateral eye (lines 188-191). I don’t think the evidence is currently very convincing about this structure actually being an eye. My doubt is because of the later description of similar dark colored carbonaceous material running along the body, described on lines 207 to 212, and suggested to be some sort of internal structure (gut, circulatory system, decay fluids, etc.). Certainly if you look only at the counterpart (UMNH.IP.6162b) in Figure 2, then you only see the dark material in the body trunk region, BUT critically there is no eye preserved in this counterpart. The “eye” is only seen in the part, UMNH.IP.6162a, and in the image of the whole part specimen (Figure 1A), the black patch called the eye lines up very well with the rest of the black material that passes along the entire length of the animal. Could the “eye” actually just be a continuation of this black material, whatever it is, into the head? Indeed, the outline of the eye is irregular and there is an extension of material in the top right of Figure 3B that doesn’t look like a typical eye structure. So the eye has the same preservation style as the rest of the black material, it is positioned perfectly in line with the other black material, the shape and outline are irregular, and eyes have never been seen in any other specimen of this genus. To me, this means that the “eye” structure is probably not an eye. The supposed “ommatidia” (line 191) would indeed give strong evidence for it being an eye, but they are not adequately shown in the photograph in figure 3B, where similar closely-packed round structures of the same size can also be seen in the orange material comprising the top left of that photo… so perhaps they are not ommatidia, but some sort of underlying structure in the sediment (sediment grains or surficial texture). Given the lack of an associated drawing of any of the specimens, and in particular the head of the part (see my earlier point), the eye is not convincing and should be re-assessed.

2. Lines 78-81: This is a nice analysis of the number of taxa known from 3 or fewer specimens, but the reporting is awkwardly phrased and needs rewording. By saying 20 taxa are known from only one specimen, and then 29 taxa are known from fewer than 3, it sounds as though you are talking about at least 49 taxa (but the total is given as 45). I suggest rephrasing for clarity to “…20 of the 45 soft-bodied species it contains (44.4%) are described in the scientific literature based on a single fossil, and an additional 9 species (another 20%) are known from only two or three specimens.” Note the use of exact percentages as well here.

3. The endites of the endopods of the first post-antennal appendages are referred to as “paired” in the caption to figure 3C, but they are not described as paired in the text (lines 194-197). Please describe why you think they are paired and include a drawing of the specimen that shows this. It is not at all clear from the photograph in figure 3C that it is paired.

5. For the exopod description, it is a bit unclear what constitutes the limits of the “slender proximal part” (line 199), versus the “paddle-shaped distal lobe fringed with robust setae” (line 199), in terms of exactly where the authors would draw the lines between these regions, and if there is an articulation between them. Also, it is unclear where they attached to the body, or how far up onto the body they extend. Again, drawings of the specimen would resolve this uncertainty. If I understand correctly, there is a prominent kink, or change in direction between the narrow proximal region and the distal lobe with setae, but this kink or elbow is not mentioned in the descriptive text. Please add it.

6. When talking about the anterior sclerite, which the authors suggest is not actually present in Emeraldella, there is mention of a single specimen for each species showing putatively this structure (line 240). In the text this is only examined in detail for E. brocki. For the pre-hypostomal sclerite of the holotype of E. brutoni described by Stein et la. 2011 and shown in their figure 2.1 and 2.2, how do the authors interpret this structure?

Additional comments

Minor suggestions for improvements to the language follow:

Line 26: By saying that the “type material” of E. brutoni consists of a single specimen, it leaves the reader wondering if there are other specimens published, which were not considered as type specimens. Maybe rephrase to something like “However, E. brutoni was described from a single specimen preserved in dorsal view…”

Line 30: Remove “main” from before “body”, because otherwise it sounds like you are referring to the trunk, rather than the whole body.

Line 39: Correct “improvise” to “improves”.

Line 48: Correct to “…opening of the first quarry…”

Line 64: Remove “Already then,” and start sentence “Most of the remarkable fossils…”

Line 77: Change to “…in the area are still officially known only from a…” (add only)

Line 93: What do you mean when you say that the deep-water deposits are “at odd” with the other sedimentation? They are highly contrasted? They are enigmatic in some way? Be more specific here (or at least use the correct term, which should be “at odds”).

Line 112: correct to “…Wheeler Formation is twice as thick…”

Line 151: correct to “…endopod podomeres vary strongly along body.”

Lines 179: replace “slopes” with “sloped”

Line 241: Correct to “…specimens of E. brocki that are exposed…”

Lines 240-241: The correct figure reference in Stein et al. 2011 for images of the pre-hypostomal sclerite are Figure 2.1 and 2.2.

Line 271: The use of “Moreover…” at the start of the sentence here makes it sounds like you are continuing a point about the antennae, when actually you are starting to describe a new difference. Rephrase for clarity.

Line 306: The use of “…shed light on this question” is a bit vague here, especially as you use it right after describing a single long held interpretation, rather than a question or conflict. Rephrase to be more specific.

Line 328: Start this paragraph with something like “In this evolutionary scenario, Emeraldella shows the most ancestral condition of all known vicissicaudates with regard to…”

Line 369: Remove “cruelly” as it is not necessary to say (even if it is true)

Line 395: Here you start a sentence with “A similar feature…” but it is unclear to which feature you are referring.

Reviewer 3 ·

Basic reporting

The manuscript presents a new specimen of Emeraldella brutoni, an arthropod from the early Cambrian of Utah. The new specimen is preserved in lateral view compared to the dorsal view on the holotype, and as such preserves much of the ventral morphology that was previously unknown from the species.

The manuscript is clear and unambiguous, with appropriate reference to the literature and suitable structure.

Experimental design

No comment.

Validity of the findings

The findings are supported by the available data, and the conclusions are well stated.

Additional comments

This is a solid manuscript, and I only have a couple of suggestions. The first is that it may be worthwhile to also re-figure the holotype, so that both specimens can be directly compared. Secondly, the preservation on the specimen does not appear to be such that you can state with certainty that there is only one articulation on the telson; the antennae, which are multiannulate, in places do not preserve the articulations and look fairly straight. The interpretation of articulations in the telson therefore should be treated with caution.

Otherwise, just a couple of typographic errors to correct:

Line 39: Presumably you mean 'improve' rather than 'improvise'.
Line 76: "End up in" rather than "End in"
Line 112: "Twice as thick" rather than "twice thicker"
Line 241: "of E. brocki that are exposed" instead of "of that E. brocki are exposed"
Line 362: "Sturmer" instead of "Surmer".

---

## Round 0.2 · accepted · Accept

I appreciate that the authors have done a very good job incorporating the comments from reviewers.